# Barriers and facilitators to accessing tuberculosis care in Nepal: a qualitative study to inform the design of a socioeconomic support intervention

Kritika Dixit [iD] ,[1,2] Olivia Biermann [iD] ,[2] Bhola Rai,[1] Tara Prasad Aryal,[1] Gokul Mishra,[1,3] Noemia Teixeira de Siqueira-Filha [iD] ,[3,4] Puskar Raj Paudel,[1,5] Ram Narayan Pandit,[1] Manoj Kumar Sah,[1] Govinda Majhi,[1] Jens Levy,[5] Job van Rest,[5] Suman Chandra Gurung,[1,3] Raghu Dhital,[1] Knut Lönnroth,[2] S Bertel Squire,[3,6] Maxine Caws,[1,3] Kristi Sidney,[2] Tom Wingfield [iD] [2,3,6]

KS and TW are joint senior authors.

For numbered affiliations see end of article.

**Correspondence to**
Dr Tom Wingfield;
tom.wingfield@lstmed.ac.uk

## ABSTRACT

**Objective** Psychosocial and economic (socioeconomic) barriers, including poverty, stigma and catastrophic costs, impede access to tuberculosis (TB) services in low-income countries. We aimed to characterise the socioeconomic barriers and facilitators of accessing TB services in Nepal to inform the design of a locally appropriate socioeconomic support intervention for TB-affected households.

**Design** From August 2018 to July 2019, we conducted an exploratory qualitative study consisting of semistructured focus group discussions (FGDs) with purposively selected multisectoral stakeholders. The data were managed in NVivo V.12, coded by consensus and analysed thematically.

**Setting** The study was conducted in four districts, Makwanpur, Chitwan, Dhanusha and Mahottari, which have a high prevalence of poverty and TB.

**Participants** Seven FGDs were conducted with 54 in-country stakeholders, grouped by stakeholders, including people with TB (n=21), community stakeholders (n=13) and multidisciplinary TB healthcare professionals (n=20) from the National TB Programme.

**Results** The perceived socioeconomic barriers to accessing TB services were: inadequate TB knowledge and advocacy; high food and transportation costs; income loss and stigma. The perceived facilitators to accessing TB care and services were: enhanced championing and awareness-raising about TB and TB services; social protection including health insurance; cash, vouchers and/ or nutritional allowance to cover food and travel costs; and psychosocial support and counselling integrated with existing adherence counselling from the National TB Programme.

**Conclusion** These results suggest that support interventions that integrate TB education, psychosocial counselling and expand on existing cash transfer schemes would be locally appropriate and could address the socioeconomic barriers to accessing and engaging with TB services faced by TB-affected households in Nepal. The findings have been used to inform the design of a socioeconomic support intervention for TB-affected households. The acceptability, feasibility and impact of this intervention on TB-related costs, stigma and TB

## Strengths and limitations of this study

► The focus group discussions contributed to new knowledge on optimal local strategies to mitigate the socioeconomic impact of tuberculosis (TB).
► The evidence has directly informed the design of a novel socioeconomic support intervention for TB-affected households, which is undergoing pilot evaluation in Nepal.
► The credibility and trustworthiness of the study was maintained through member checking, using multiple coders, conducting a consensus-based coding, recruiting local interviewers for data collection, performing triangulation and including a broad selection of multidisciplinary stakeholders to inform the study conclusion.
► The study was conducted in four districts of Nepal, mostly lowland 'plains' districts, which could affect the transferability of the findings.
► People who were diagnosed with TB in private sectors or those lost to follow-up did not participate in the study despite, in other settings, having been shown to be groups at high risk of severe socioeconomic impact of TB.

treatment outcomes, is now being evaluated in a pilot implementation study in Nepal.

## BACKGROUND

Tuberculosis (TB) kills 1.3 million people each year worldwide, more than any other single infectious disease including, up to the time of writing, COVID-19.[1] In 2019, an estimated 10 million became ill with TB, of whom 2.9 million were not notified or remained undiagnosed and untreated.[1] In low-income and middle-income countries (LMICs), stigma, marginalisation and catastrophic costs of accessing TB diagnosis and

care, coupled with limited social protection coverage, can delay diagnosis, decrease TB treatment success rates and push TB-affected households into further impoverishment.[2 3] To address this and move towards TB elimination, the WHO's (WHO) 2015 End TB Strategy set the bold target that 'Zero TB-affected families should face catastrophic costs' and that psychosocial and economic (socioeconomic) support should be provided to TB-affected people.[4]

Nepal is an LMIC in South Asia with significant TB incidence (annual incidence 245/100 000) and mortality.[5] Despite free basic TB diagnostic tests, medicines and financial support for people with drug-resistant (DR-TB), approximately one in two people with TB face catastrophic costs (defined as the total TB-related costs equivalent to greater than 20% of a household's annual income) while accessing TB care in Nepal.[6 7] Such costs include travel for directly observed treatment short-course (DOTS), additional food expenditure and lost income, which can contribute to adverse TB treatment outcomes, especially for the poorest, most vulnerable households.[6 8–10]

The Nepal National TB Programme (NTP) provides NPR3000 (~US$27) in cash incentives monthly for transportation and nutritional support to people with multidrug-resistant TB (MDR-TB)[11] who are enrolled in government treatment centres and receiving ambulatory care. There is currently no cash incentive scheme for people with drug-sensitive TB (DS-TB) in Nepal.[12] In other settings, socioeconomic support for TB-affected households, including mutual support groups and cash transfers, has been shown to help overcome barriers to accessing TB services, defraying catastrophic costs and improving treatment success rates.[3 13–16] However, there is limited context-specific understanding of the barriers and facilitators to TB diagnosis and care in LMICs with which to inform the development of tailored socioeconomic support interventions for people with TB and their households.[1] This study aimed to address this knowledge gap in Nepal.

## MATERIALS AND METHODS
### Study design
We conducted an exploratory qualitative study, which used semistructured focus group discussions (FGDs) to collate the perceptions of key stakeholders in Nepal regarding socioeconomic barriers and facilitators of accessing and engaging with TB diagnosis and care. The study formed part of a larger programme of mixed-methods research[17] to design a locally appropriate socioeconomic support intervention for TB-affected households. The study adhered to the COnsolidated criteria for REporting Qualitative research (COREQ) Checklist.[18]

### Study setting
The study was conducted in four districts of Nepal where Birat Nepal Medical Trust (BNMT), a Nepalese organisation with a focus on TB-related implementation

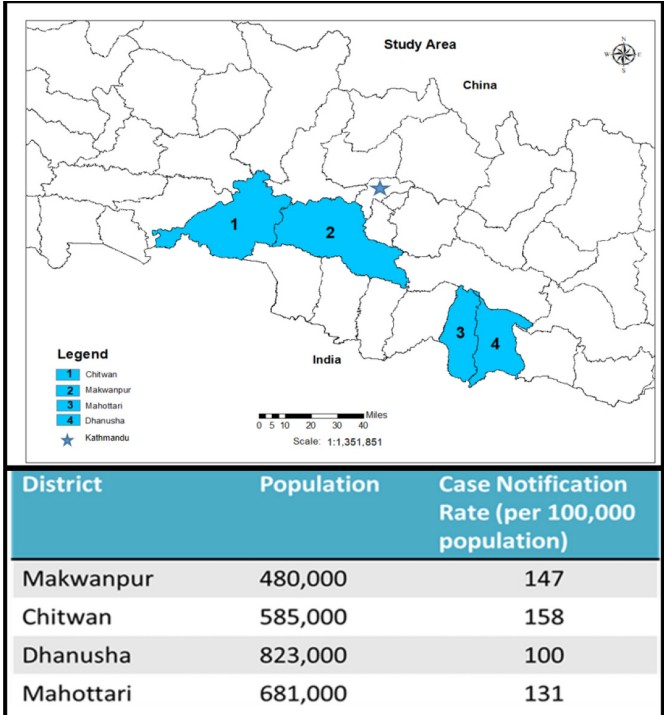

| District | Population | Case Notification Rate (per 100,000 population) |
|---|---|---|
| Makwanpur | 480,000 | 147 |
| Chitwan | 585,000 | 158 |
| Dhanusha | 823,000 | 100 |
| Mahottari | 681,000 | 131 |

**Figure 1** The highlighted colour represents the study districts in Nepal. Dhanusha, Mahottari and Chitwan are 'plains' or 'Terai' districts. Makwanpur is a hilly district. The district's data for population numbers and TB case notification rate highlight the burden of tuberculosis in each district (National TB Control Center Annual Report, 2018). TB, tuberculosis.

research, implemented IMPACT-TB project. The districts: Makwanpur, Chitwan, Dhanusha and Mahottari have a high prevalence of poverty and TB (figure 1).[11] Makwanpur is a hilly district with limited road networks. Other three districts are lowland plains and challenged by high population density, poor health indicators and high rates of illiteracy.

### Sampling
A desk-based scoping exercise was initially performed by team members (KD, RD and TW) to identify participants from relevant stakeholder groups in Nepal. To collate diverse perspectives on barriers and facilitators of TB diagnosis and treatment, the team purposively selected participants who had direct or indirect experiences with TB services. The participants included: people affected by TB who were currently receiving or had recently completed DS-TB or MDR-TB treatment with the NTP (referred to as 'people with TB' in the study); community leaders such as female community health volunteers, teachers and social leaders from civil society organisations (CSOs, referred to as 'community stakeholders'); and TB healthcare professionals, including those working with the NTP, community volunteers and TB-focused non-governmental organizations (NGOs) (referred to as 'NTP stakeholders') (table 1). The list of people with TB, including their demographics, were gathered from the

**Table 1** List of FGD stakeholder groups and participants

| Stakeholder group | Sex | Age group (years) | District | Total no of participants |
|---|---|---|---|---|
| People diagnosed with TB, mixed sex group | Female | Under 20 | Makwanpur | 7 |
| | Male | 25–30 | Mahottari | |
| | Male | 45–50 | Makwanpur | |
| | Male | 20–25 | Dhanusha | |
| | Female | Under 20 | Makwanpur | |
| | Male | 30–35 | Chitwan | |
| | Male | 55–60 | Chitwan | |
| People diagnosed with MDR-TB | Male | 40–45 | Chitwan | 7 |
| | Male | 70–75 | Chitwan | |
| | Male | 20–25 | Chitwan | |
| | Male | 20–25 | Chitwan | |
| | Male | 45–50 | Chitwan | |
| | Male | 45–50 | Chitwan | |
| | Female | 20–25 | Chitwan | |
| Females diagnosed with TB | Female | 60–65 | Mahottari | 7 |
| | Female | 25–30 | Makwanpur | |
| | Female | 40–45 | Mahottari | |
| | Female | 45–50 | Chitwan | |
| | Female | 45–50 | Dhanusha | |
| | Female | 25–30 | Dhanusha | |
| | Female | 25–30 | Makwanpur | |
| Community leaders | Female | 50–55 | Chitwan | 6 |
| | Male | 45–50 | Mahottari | |
| | Male | 35–40 | Makwanpur | |
| | Male | 45–50 | Chitwan | |
| | Female | 50–55 | Chitwan | |
| | Male | 40–45 | Dhanusha | |
| Civil society organisation | Male | 35–40 | Chitwan | 7 |
| | Male | 40–45 | Chitwan | |
| | Male | 65–70 | Chitwan | |
| | Male | 25–30 | Mahottari | |
| | Male | 45–50 | Chitwan | |
| | Male | 45–50 | Makwanpur | |
| | Male | 25–30 | Dhanusha | |
| TB healthcare professionals | Male | 55–60 | Kathmandu | 12 |
| | Male | 30–35 | Kathmandu | |
| | Male | 30–35 | Kathmandu | |
| | Male | 30–35 | Kathmandu | |
| | Female | 25–30 | Kathmandu | |
| | Male | 55–60 | Kathmandu | |
| | Male | 45–50 | Kathmandu | |
| | Male | 45–50 | Kathmandu | |
| | Male | 55–60 | Kathmandu | |
| | Male | 55–60 | Kathmandu | |

Continued

 

**Table 1** Continued

| Stakeholder group | Sex | Age group (years) | District | Total no of participants |
|---|---|---|---|---|
| | Male | 55–60 | Kathmandu | |
| | Male | 45–50 | Kathmandu | |
| Community mobilisers | Male | 45–50 | Dhanusha | 8 |
| | Female | 30–35 | Chitwan | |
| | Female | 30–35 | Makwanpur | |
| | Male | 25–30 | Dhanusha | |
| | Male | 30–35 | Chitwan | |
| | Male | 40–45 | Mahottari | |
| | Male | 25–30 | Mahottari | |
| | Female | 20–25 | Makwanpur | |
| Total | Male: 38 Female: 16 | | | 54 |

Weaver et al[22].
FGD, focus group discussion; TB, tuberculosis.

IMPACT TB database or registers of the health clinics in each district. Community stakeholders were community leaders or those working in civil society and were selected based on their in-depth knowledge on the local context and experiences of working with the communities, preferable in health progammes. TB healthcare professionals, such as those working with the NTP or TB-focused NGOs, have several years' experience in planning, designing and implementing NTP activities. Community volunteers or mobilisers were people working with the IMPACT-TB project, who have first-hand experiences in screening symptoms of TB and supporting people with TB to adhere to and complete their treatment. These participants were selected based on the expertise in delivering community programmes and to bring diverse perception of the stakeholders into the study. Using telephone, email or in-person meetings, we invited 55 individuals to participate in the study. Inclusion criteria were being 18 years of age or older and being able to give informed consent. Seven participants were invited to each of the seven FGDs with the exception of the TB healthcare professional FGD, which consisted of 12 participants. This related to the logistical challenges of organising more than one FGD with this group due to their working hours and time constraints coupled with the aim of representation from the public, private and NGO sectors of TB healthcare.

### Data collection

The study team consisted of diverse members from multiple sectors including a physician, senior TB researchers, social scientists, public health professionals and project managers. An interview guide was developed by the coauthors with previous qualitative methods experience: TW (male, principal investigator, TB researcher), KD (female, doctoral student, project manager) and BR (male, public health specialist, research associate); BR and KD are employed by BNMT. The interview guide consisted of open-ended questions to explore the perceptions of participants concerning protective factors and risk factors for exposure to TB and development of TB disease; barriers and facilitators to accessing and engaging with TB diagnosis and care, including the recommendations for better access and engagement with TB diagnosis and care; and the socioeconomic impact on people with TB of being ill with the disease.

Prior to conducting FGDs, participants were provided with a 'Participant Information Sheet' that explained the purpose of the study, benefit and harm, and confidentiality.[17] Participants were provided time as they would require to read and understand the information in the paper and then decide if they are willing to participate in the study. The FGDs were conducted in a local hotel accessible to participants in the study districts. The topic guide was piloted with a group of seven female and male participants with TB resulting in minor refinements to the FGD structure and delivery techniques. TW moderated the FGD with TB healthcare professional and KD and BR moderated the other six FGDs. Apart from these researchers, district field staff who supported patients attended the discussions and facilitated any dialectic interpretation or contextual explanations related to access to and engagement with TB services.

We conducted seven FGDs with 54 participants, which the project team perceived as giving sufficient information power for the study.[19] Of the participants, three-quarters were male and the average age was 42 years (table 1). To encourage an environment in which participants felt comfortable and able to share their opinions and to balance gender representation, two of these FGDs were specifically for females with DS-TB and female TB community mobilisers. In all the FGDs, there were seven participants, except for the FGD with community leaders (n=6), FGD with TB healthcare professionals (n=12) and

FGD with community mobilisers (n=8). One invitee from the community leader's FGD declined to participate due to lack of time. One additional community mobiliser showed interest to participate in the FGD with community mobilisers' and was also included. We did not conduct any follow-up discussions with participants but some of the participants attended a workshop to discuss the FGD findings, the outputs of which are published elsewhere.[12] We performed real-time member checking in each FGD by noting key points of the discussion, summarising them on a wall chart and clarifying their accuracy with the group. No formal field notes were taken. The FGDs, which lasted 90–120 min, were all conducted in Nepali language apart from the FGD with TB healthcare professionals, which was conducted in English. FGDs were audiorecorded, translated into English from Nepali language and back-translated by an independent translator who was not part of the project team. Each FGD was concluded when the facilitators collectively felt the topics in the FGD interview guide were sufficiently explored.

## Analysis

We applied thematic analysis using NVivo V.12 to manage the data.[20] The study used multiple coders, KD and TW, who familiarised themselves with the data through successive reading of transcripts. KD and TW separately generated the initial codes for each transcript before discussing and comparing the perception of understanding of the codes. The codes were updated through regular discussion as further data became available and collated following each successive FGD. To increase trustworthiness of the study, after all the transcripts were coded and analysed, KD and TW independently reviewed coding and themes and refined them through further discussion, triangulation and consensus where necessary.[21] Both open and closed first-order categories were used to label data. Categories were then grouped into second-order and third-order themes (online supplemental file 1). Table 2 shows an example of the analysis process of codes and themes. To better inform the design and delivery of the socioeconomic intervention for TB-affected households within the wider context of health services delivery, the first-order themes were then mapped to four levels of an adapted WHO Treatment Adherence Framework: (1)

TB, health and basic education; (2) social protection and nutrition; (3) psychosocial and (4) healthcare system, TB diagnosis and care delivery.[22] We chose to structure our analysis on the themes mapped to levels 1–4 of the WHO Framework because these levels were the most relevant to the study's aim of informing design and development of a socioeconomic support intervention for TB-affected households. While important, themes identified that mapped to category IV of the Framework, such as governmental policy, political commitment, public–private mix and healthcare infrastructure were perceived by the study team to be largely unmodifiable by a household-level socioeconomic support intervention. These themes are reported under health system categories and are shown in online supplemental file 2.

The study protocol is provided in online supplemental file 3. Written consent was obtained from all participants. Confidentiality of the participants was maintained by anonymising FGD responses, keeping any paper data in a locked cabinet at BNMT's office and securing the data in a password-protected database.

### Patient and public involvement

Patient and/or the public were not involved in the design or conduct of this research.

## RESULTS

Overall, 36 codes related to eight themes were identified (online supplemental file 1). Below, we focus only on the perceived barriers and facilitators that mapped to categories: (1) TB, health and basic education, (2) social protection and nutrition and (3) psychosocial, of the adapted WHO Framework. These barriers and facilitators are shown in figure 2.

### TB, health and basic education
#### Theme: information barrier to access and adhere to TB diagnosis and treatment

Across FGDs, we identified low literacy and education about TB as a barrier to accessing TB diagnosis and engaging with TB treatment (figure 2). Knowledge about TB including transmission, prevention, symptoms how and where to get diagnosed, treatment regimens and

| Table 2 | An example of coding from the FGDs | | | | |
|---|---|---|---|---|
| **FGD** | **Quote** | **First-order category*** | **Second-order themes** | **Third-order themes** |
| People diagnosed with TB | FGD with people diagnosed with TB, 30–35 years age group, male: 'People get criticized for having TB. The community perceives a TB patient isn't the same as a normal person…. due to lack of awareness.' | Psychosocial | ► Enacted stigma<br>► Perceived stigma<br>► Lack of knowledge | Stigma as social barrier to access |

*Adapted from a WHO Medication Adherence Framework.[22]
FGD, focus group discussion; TB, tuberculosis.

 

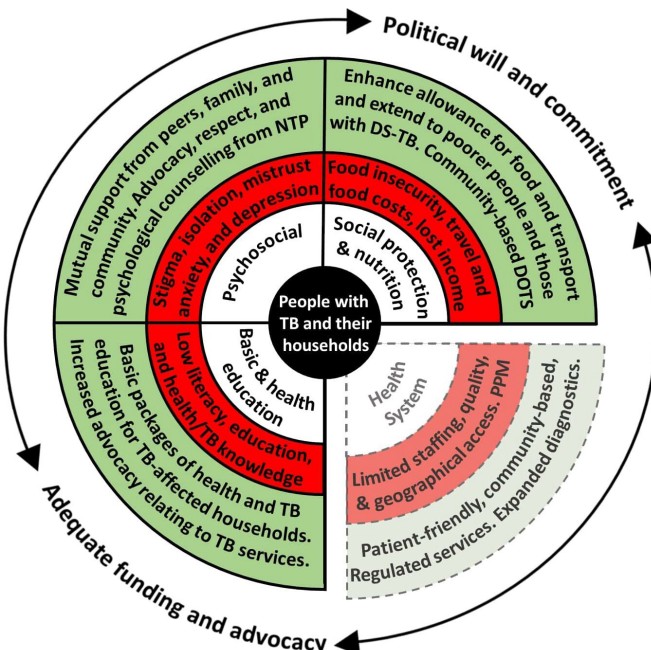

**Figure 2** The inner white circle contains the key categories that influence tuberculosis (TB) service access and engagement, which are adapted from a WHO medication adherence framework (see the Methods section).[22] The middle red circle indicates the main barriers identified for each category, which may threaten access to TB services. The outer green circle indicates the main facilitators (current or potential) for each category, which may enhance access to TB services. Barriers relating to 'TB, health and basic education', 'social protection and nutrition', and 'psychosocial' were perceived by the project team to be modifiable by a household level socioeconomic intervention. Barriers relating to the 'health system' were perceived by the project team to be non-modifiable by a household-level socioeconomic intervention and are, therefore, separated from the other categories and represented by dotted lines. 'PPM' as a health system barrier refers to the protracted and convoluted patient journey through public and private healthcare providers, which was reported as being associated with increased economic impact, especially related to out-of-pocket costs. The surrounding bidirectional arrows indicate the cross-FGD finding that adequate funding and advocacy, and political will and commitment were perceived as vital structural factors to enable the facilitators identified to overcome the barriers identified. DOTS, directly observed treatment short-course; DS-TB, drug-sensitive tuberculosis; FGD, focus group discussion; NTP, National Tuberculosis Programme; PPM, public–private mix. Reference: Weaver et al[22].

duration, and the TB services available at health facilities were perceived to be limited, especially among illiterate and underserved groups or rural populations. A female participant with TB said: *'I didn't know that TB medicines were free in hospitals. I knew it only when I visited the health post.'*

The FGD with community stakeholders suggested that this limited knowledge about TB negatively impacted engagement with TB services and treatment adherence. This was noted to potentially increase the likelihood of

a delayed diagnosis, more advanced disease at presentation and acquisition of DR-TB during treatment. The perceived lack of knowledge was predominantly felt to relate to suboptimal TB education, advocacy and political commitment from health system and governmental leaders. Participants also reported that the current health education programmes are scarce and unable to reach poorer, educationally and socially marginalised communities with high TB risk.

### Theme: facilitating treatment access through education about TB disease and advocacy about TB services, especially in remote communities

Across all the FGDs, participants described the need to raise community awareness on risk factors for TB, mode of TB transmission, and TB signs and symptoms. Community stakeholders particularly felt that future awareness-raising programmes would benefit by informing communities about available free TB diagnostic and treatment services in their local area and adding a component to reduce TB-related stigma. Further, NTP stakeholders stressed the importance of not only providing education but also, in the face of competing health beliefs, influencing attitudes and promoting behavioural change.

Community stakeholders suggested that the government should take responsibility for the development and implementation of intensive household-level and village-level awareness programmes using technology such as smartphone applications and social media. They also recommended broadcasting educational campaigns on television adapted from similar campaigns in the field of HIV (HIV ka sawal) and maternal care (Aama ko Maya) in Nepal. However, there was concern that information transmitted through media could bypass vulnerable, poor and marginalised populations. To overcome this, the group suggested innovative and interactive community-based approaches such as coordination with local women's group-initiated self-help enterprises, street plays and engaging TB survivors as peer champions and educators to improve TB education. The involvement and ownership of TB education programmes by community leaders, including volunteers, teachers and community mobilisers, was also deemed important to achieve effective, decentralised delivery. Furthermore, the stakeholders acknowledged the essential role of healthcare provider-led education as part of a client–provider contract because people with TB need—and will follow—advice from healthcare providers only when that advice is relayed sensitively and understood thoroughly. Nevertheless, people with TB indicated that both sensitisation and education from a trusted source were key to deliver education successfully and to enable and empower communities.

FGD with community leaders, 50–55 years age group, female: *'We need to create [educational] groups attached to health centres and schools. Community and locally-elected leaders and teachers could give education to their communities and conduct TB awareness training and workshops.'*

## Social protection and nutrition
### Theme: social and economic barriers to accessing and engaging with TB diagnosis and treatment

Across FGDs, the participants reported food insecurity, high travel and food costs, and lost income as key barriers to accessing and engaging with TB diagnosis and treatment (figure 2). The direct out-of-pocket costs of seeking TB diagnosis and engaging with DOTS at both public and private clinics, including food and transport, were raised repeatedly across FGDs as a significant barrier to timely diagnosis and medication adherence.

FGD with community leaders, 50-55 years age group, female: *'TB medicines are free but people also need money for two-way transportation and food. TB illness [and even] TB treatment can make people weak and nutrition is needed. How can people afford these [nutrition and transportation] costs?'*

Nonetheless, patient journeys were repeatedly reported as long and convoluted, including a public–private mix of traditional medicine, pharmacies, local private healthcare providers and larger private clinics before reaching NTP diagnostic and treatment services (figure 2). As a result, TB-affected households incurred significant expenses.

FGD with people diagnosed with TB, 45-50 years age group, male: *'I visited all the pharmacies in my city, about 15–17 medicals [pharmacies] overall. I used to buy pneumonia medicine and take it every time. TB was not initially identified. After a month without taking any medicines, TB was identified [at the government clinic]… The barrier is more financial. I spent approximately 1.7 lakhs [~1530 USD] for my treatment.'*

Undernutrition and food insecurity were recognised as risk factors for TB and to contribute to suboptimal adherence. Across FGDs, it was noted that many TB-affected households have insufficient resources to buy nutritious food. In addition, it was perceived that lack of education negatively impacted decision making related to nutrition to recover during TB treatment. In particular, community leaders and people with DR-TB raised concerns that the current monthly governmental nutrition allowance was insufficient to obtain nutritional food, such as meat and fruits, which is often recommended by healthcare providers. Female participants with TB mentioned a trade-off between out-of-pocket expenses to purchase the recommended nutritional food vs transportation expenses associated with DOTS at TB clinics.

Lost income was identified in the majority of FGDs as a notable barrier to accessing diagnosis and engaging with TB care, which compounded the economic impact of out-of-pocket travel and food costs. People with TB expressed guilt and stress related to having to stop work following their TB diagnosis and the fear of not getting employment again. Female participants described the impact as hardest on poorer populations, women working in the fields, and labourers, who lack the free time required to go to the health facility for daily DOTS, especially when opening hours and prime labouring hours coincide.

Community stakeholders and people with TB frequently cited that TB-affected households use coping strategies to mitigate the economic impact of TB, most commonly to obtain funds to buy food. Selling assets, such as livestock, milk, land and jewellery, was reported as a predominant coping strategy. However, it was noted that some vulnerable patients were so poor that they have nothing to sell. Other coping strategies mentioned included borrowing money, formally and informally, which led to further economic hardship and difficulties maintaining adherence.

FGD with people diagnosed with TB, 25-30 years age group, male: *'I had difficulties [to pay money to access TB services] and wasn't able to go [to the clinic] for a month. I had to borrow money from my friends.'*

### Theme: facilitating treatment adherence by people with TB through nutritional and/or economic support

Participants discussed the need for social protection including insurance, transportation allowance and nutritional support for TB-affected people (figure 2).

The potential for economic support to improve nutrition, defray travel and other out-of-pocket costs, and increase TB treatment success, was raised in all FGDs. It was acknowledged, particularly in FGDs with NTP stakeholders and people with TB, that the government of Nepal provides Rs 3000/month (~US$ 27 for ambulatory MDR-TB cases and Rs 1000/month (~US$8) for those staying at DR-TB hostels. While the NTP stipulates that this is intended as nutritional and/or transport allowance, participants noted that how the cash is spent is not monitored. In addition, issues were raised with this existing transfer scheme, including delays in delivery of the allowance.

FGD with CSO, 45-50 years age group, male: *'The most important question is when to give the allowance. It would be better in the first phase [of treatment] because it is [most] valuable at this time when one needs it most.'*

There was further debate concerning whether cash or nutritional support was most appropriate. Some community stakeholders raised concerns regarding misuse of cash payments (eg, to buy alcohol) and suggested that it would be preferable to provide nutritious food such as milk, ghee (local butter), meat and eggs. However, females with TB perceived that any cash received would nevertheless be spent on food, primarily staple foods such as rice, to feed their household.

Finally, it was perceived across FGDs that any nutritional or economic support should either be provided to all or stratified by need rather than TB drug resistance profile.

FGD with people diagnosed with TB, 30-35 years age group, male: *'The government should provide nutritious food based on the economic status of patients. Drug-sensitive patients should also be provided with an allowance based on their level of poverty.'*

## Psychosocial
### Theme: psychosocial barriers to treatment adherence

Across all FGDs, stigma was perceived to be a significant barrier to seeking, accessing, and engaging with TB diagnostic and treatment services. People with TB

described feeling discriminated against, mistreated, isolated or hated. They reported perceptions or instances of people talking behind their back or remaining physically distanced. Participants of the FGD with community stakeholders shared that sometimes people with TB experienced extreme negative behaviour such as physical or psychological mistreatment from their own family members. The situation was discussed as being even more pronounced for young married females because of a lack of personal agency within their husband's family. While this stigmatising behaviour towards people with TB was reported to occur across socioeconomic groups, participants described that a 'blame and shame culture' was prevalent among family members belonging to groups perceived as 'higher' in the caste related, social hierarchy. In alignment with this assertion, community mobilisers mentioned that lack of social and family support can cause people to conceal their TB status and not adhere to or complete TB treatment.

Reports of stigmatising behaviour were not limited to the community and family members. Perceived negative behaviour of healthcare providers towards people with TB was noted across FGDs as an issue that compounded self-stigma and led to a breakdown of trust within the client–provider relationship. Although participants felt that, generally, enacted stigma had decreased in Nepalese communities, FGDs with people with TB and community stakeholders shared that people still fear TB disease, especially in rural villages.

FGD with people diagnosed with TB, 30-35 years age group, male: *'People may know they have symptoms of TB but are too ashamed to go to the health facilities. People can't say out loud that they have TB. TB is regarded as a big disease and people get criticized for having it. The community perceives a TB patient differently than a normal person due to lack of awareness. That's why it's difficult to end TB.'*

People with MDR-TB reported profound psychosocial impact including anxiety and isolation, especially during the first months of treatment. Depression, suicidal ideation and shame related to stigma and also well-recognised side effects of MDR-TB medications were mentioned.[23]

FGD with people diagnosed with MDR-TB, 45-50 years age group, male: *'I wanted to die. One of my friends [with MDR-TB] committed suicide after 16 months [of treatment].'*

### Theme: Mutual or social support as a facilitator to treatment adherence

Social support from family and friends was perceived as a facilitator to adhering to TB treatment and becoming cured (figure 2). This included visiting, spending time with and showing affection towards people with TB to demonstrate solidarity and reduce feelings of isolation.

People with TB also shared the importance of mutual support beyond family and friends, including the wider community, leaders, elders and other important local figures. The participants believed that this kind of support would help people with TB to cope and reflected the

close communities and rich socio-cultural values inherent to Nepalese culture.

FGD with people diagnosed with MDR-TB, 40-45 years age group, male: *'My friends and the people in my village told me 'TB is a normal disease and encouraged me that, if I took my medicine, I'd be alright.'*

Interactions and consultations with healthcare providers were also seen as opportune occasions to provide education and counselling to address the psychological impact of TB. FGD with community stakeholders raised the perceived importance of healthcare providers simply recognising, acknowledging and being understanding of patients' fears, concerns and expectations. While healthcare provider-led counselling on medication adherence was noted to be commonplace at treatment initiation, counselling patients with TB about TB-related fear, stigma, depression and anxiety was broadly overlooked. It was noted that integration of medication and psychological counselling by healthcare providers could be a suitable method to deliver clear and open information about stigma and discrimination, which could improve TB treatment adherence and completion rates and potentially support mental wellness and empowerment.

## DISCUSSION

This qualitative study generated new evidence regarding barriers and facilitators to accessing and engaging with TB services in Nepal. Multisectoral stakeholder participants highlighted that the barriers were predominantly related to the poor socioeconomic conditions of people with TB, including lack of education and endemic poverty. The findings showed that the costs of care-seeking and clinic-based DOTS can further compound poverty and, when combined with psychological impacts including stigma and anxiety, were perceived to negatively influence access to TB services. Participants cited multiple potential socioeconomic interventions, both integrated and discrete, including TB education, economic, nutritional and social support, to mitigate catastrophic costs of TB-affected households and support people with TB to get cured.

### Knowledge and awareness about TB

Low education levels and limited awareness of TB are associated with delays in healthcare seeking.[24–26] A study in Nepal showed inadequate knowledge of TB was associated with increased likelihood of consulting traditional healers, resulting in TB diagnostic delay.[27] Our findings are also similar to other studies that suggested knowledge about TB was limited in poor, marginalised and/or rural communities in Nepal.[8 28] This implies that any existing TB education and advocacy programmes may not be reaching crucial, high-risk target groups and new approaches are required if Nepal is to end TB.

Educational support interventions that enhance knowledge about TB transmission, symptoms, treatment and prevention, are important contributory factors in both care-seeking behaviour and treatment outcomes

in diverse settings.[27 29 30] In India, the Global Fund-supported advocacy, communication and social mobilisation project, 'Axshya', has made progress towards reaching underserved groups through intense community outreach and education.[31] FGD participants in our study cited a dearth of awareness-raising interventions and campaigns in recent years in Nepal. Previously, similar campaigns focused on TB awareness through door-to-door visits, health promotion at health facilities or educational outreach into communities. Such campaigns were perceived to increase knowledge on TB, advocate for free TB services and empower communities to make informed choices. As highlighted by participants, in addition to commonly used platforms such as leaflets, radio and television, future educational campaigns in Nepal would benefit from using technology such as mobile phones—which are used by over 90% households in both urban and rural areas[32]—or, where appropriate, social media.

## Psychosocial impact

Of the perceived psychosocial barriers to accessing TB diagnosis and care in Nepal, stigma predominated. Participants mentioned feelings of guilt among people with TB, fear of disclosure and experience of discrimination. This mirrors findings from diverse settings, which show that experiences of stigma are highly prevalent among people with TB and can impede access to TB services.[33] For example, in Zambia, a cohort study showed that anticipated and enacted stigma of people with TB resulted in delayed diagnosis, poor treatment adherence, reduced quality of life and represented a distinct challenge to successful screening of their household contacts.[34]

It was notable that people with MDR-TB reported severe negative psychosocial impacts of their illness. These included profound feelings of anxiety, isolation related not only to their diagnosis but also to physical distance from their families, and recognised side effects of certain MDR-TB medications (eg, cycloserine) such as depression and despair.[35–37] Participants perceived an association between the psychosocial impact of MDR-TB and the potential for non-adherence to long, arduous treatment regimens including injectable agents. Discussion across FGDs suggested that existing medication adherence counselling delivered by NTP staff at treatment initiation would be a suitable platform on which to integrate complementary psychosocial counselling about overcoming TB-related stigma and addressing ill mental health.

## Economic burden of TB

The economic impact of accessing TB diagnosis and care was perceived to be severe. This was mainly due to high costs associated with transportation to clinics, maintaining adequate nutrition and time and income loss. Participants with MDR-TB indicated that there was delay or unavailability of the NTP's financial assistance scheme during their treatment. The financial impact of belonging to a household affected by TB was cited in

FGDs as forcing households to resort to coping strategies such as taking out loans, using savings and selling assets. These findings are in line with the rapidly growing global body of evidence relating to the economic burden of TB. Such findings suggest that coping strategies remain common and only limited progress has been made towards the WHO target of 'zero TB-affected families face catastrophic costs by 2020'.[1 38] TB Patient Costs Surveys conducted in various LMICs have demonstrated that a substantial proportion of TB-affected households incur catastrophic costs, which can push them into further impoverishment and contribute to adverse TB treatment outcomes.[6 39–44] Studies demonstrated that more than 60% of TB-affected households in Nepal incurred catastrophic costs[6 7] and stark economic impact.[27 45–47]

Although this study focused on barriers amenable to interventions at the household level rather than health system level, our findings showed that when TB diagnosis and care were sought from both the public and private sector, patient pathways to TB diagnosis and care were protracted and their costs, especially out-of-pocket costs, escalated (described in detail in online supplemental file 2). This finding is consistent with the findings of systematic reviews from Nepal, India and Uganda,[48–54] which also highlighted that interventions to strengthen public–private partnerships can streamline diagnostic and referral pathways and potentially increase TB notifications to the NTP. Studies in India and Vietnam have demonstrated enhanced engagement with private pharmacies and medical practitioners by providing them subsidies directly or through intermediary agencies for every notified case.[55 56] This could be a potential strategy to improve access to TB care in Nepal, where approximately 20% people with TB receive paid treatment from the private sector.[57]

## Social and economic support

To address the psychosocial and economic impact of TB and improve TB cure and prevention rates, our findings imply a need for both social and economic support. This is supported by the results of studies in Nepal and other LMICs, which showed that providing both counselling and economic assistance to people with MDR-TB improved cure rates.[9 58–60] However, it must be noted that a significant proportion of people with DS-TB in Nepal experience enduring psychological, social and economic impacts of TB but receive no additional support.[6 7 12] In line with a study from Ethiopia,[61] our findings also suggest that the timing of provision of financial support is important. Participants advocated for early cash support delivery in the initiation phase of treatment when they perceived it to be needed most. Involvement of family members and peers in such interventions was also noted by participants as a vital aspect of support to complete TB treatment. This has also been reported in a systematic review of factors affecting medication adherence in LMICs.[62]

A well-designed socioeconomic support intervention would ideally be tailored or stratified to the individual or household needs of a person with TB and include overlapping elements such as increasing knowledge, awareness-raising, cost mitigation (eg, through cash transfers or transport vouchers) and stigma-reduction activities (eg, mutual support, peer groups, enhanced medication counselling sessions), integrated into existing TB services.[63 64] However, it must be acknowledged that such an intervention would need to balance stratification with feasibility and pragmatism.

### Strengths and limitations

Our study fills an important gap in knowledge about household-level socioeconomic barriers to accessing TB services in Nepal and expanded on perceived facilitators and enablers to overcome these barriers. A major strength of the study methods was the trustworthiness and validity harnessed by garnering perspectives of the diverse study participants.[65]

The study has several limitations. As the participants were predominantly from Terai plains districts the findings, therefore, should be cautiously applied in other settings or countries. Nevertheless, we described the study setting to improve transferability. Second, participants from NGOs and healthcare professionals working with the NTP were over-represented within the participant cohort. However, we tried to strike a suitable balance by including the views of people with TB and community groups, which historically have been overlooked in similar research. FGDs consisted of limited women participants. We minimised the issue by specifically describing the female participants' viewpoint in the analysis. Similarly, there was no participation of people diagnosed with TB in private sectors or those who had been lost to follow-up who are the vulnerable to restricted access to TB services and poor outcomes. Their engagement would enable us to fully understand the most important barriers to care.[39 64]

### CONCLUSION

There are multiple socioeconomic barriers to accessing and engaging with TB services in Nepal. TB education and advocacy, economic support and psychosocial counselling integrated with medication adherence counselling could address these barriers and potentially reduce stigma, mitigate TB-related costs and improve TB treatment outcomes. These elements are now being incorporated into the design of a locally appropriate socioeconomic support intervention for TB-affected households for pilot implementation in Nepal.

**Author affiliations**
[1]Department of Research, Birat Nepal Medical Trust (BNMT), Kathmandu, Nepal
[2]Department of Global Public Health, WHO Collaborating Centre on Tuberculosis and Social Medicine, Karolinska Institute, Stockholm, Sweden
[3]Departments of Clinical Sciences and International Public Health, Liverpool School of Tropical Medicine, Liverpool, UK
[4]Department of Health Sciences, University of York, York, UK
[5]KNCV Tuberculosis Foundation, Den Haag, The Netherlands
[6]Tropical and Infectious Disease Unit, Liverpool University Hospital NHS Foundation Trust, Liverpool, UK

**Acknowledgements** We would like to acknowledge National Tuberculosis Control Center, Ministry of Health and Population, Nepal Health Research Council, Health Directorate of Province 2 and Province 3, and all related health facilities and authorities for their support and collaboration with this project. Most importantly we are grateful to people with TB for their generosity of time and sharing their feedback and experiences during this study.

**Contributors** KD: data curation, investigation, project administration, methodology, analysis, first draft preparation, writing-review and editing. OB and KS: methodology, writing-review and editing. BR and PRP: data curation, project administration, writing-review and editing. TPA, RNP, MKS and GMa: investigation, project administration, writing-review and editing. GMi: investigation, methodology, supervision, writing-review and editing. NTdS-F: data curation, formal analysis, investigation, methodology, project administration, writing-original draft preparation, writing-review and editing. JL and JvR: data curation, resources, validation, writing-review and editing. SG and RD: investigation, project administration, resources, supervision, writing-original draft preparation, writing-review and editing. KL and BS: conceptualisation, methodology, supervision, writing-original draft preparation, writing-review and editing. MC: conceptualiSation, formal analysis, funding acquisition, investigation, methodology, project administration, supervision, writing-original draft preparation, writing-review and editing. TW: conceptualisation, data curation, formal analysis, funding acquisition, investigation, methodology, project administration, supervision, validation, visualisation, writing-original draft preparation, writing-review and editing.

**Funding** KD receives support from the Farrar Foundation and Royal Society of Tropical Medicine and Hygiene (RSTMH) and National Institute of Health Research (NIHR). TW is supported by grants from: the Wellcome Trust, UK (209075/Z/17/Z); the Medical Research Council, Department for International Development, and Wellcome Trust (Joint Global Health Trials, MR/V004832/1), the Academy of Medical Sciences, UK; and the Swedish Health Research Council, Sweden. All other authors (KD, OB, BR, TPA, GMi, GMa, NTdS-F, PRP, RNP, MKS, JL, JvR, SG, RD, KL, BS, MC and KS) are supported by EU Horizon2020 grant 733174 IMPACT TB.

**Map disclaimer** The inclusion of any map (including the depiction of any boundaries therein), or of any geographic or locational reference, does not imply the expression of any opinion whatsoever on the part of BMJ concerning the legal status of any country, territory, jurisdiction or area or of its authorities. Any such expression remains solely that of the relevant source and is not endorsed by BMJ. Maps are provided without any warranty of any kind, either express or implied.

**Competing interests** None declared.

**Patient consent for publication** Not applicable.

**Ethics approval** The study received ethical approvals from the University of Liverpool (No. 2436) and the Nepal Health Research Council (208/2018).

**Provenance and peer review** Not commissioned; externally peer reviewed.

**Data availability statement** All data relevant to the study are included in the article or uploaded as online supplemental information.

**ORCID iDs**
Kritika Dixit http://orcid.org/0000-0002-7957-8109
Olivia Biermann http://orcid.org/0000-0002-5978-0211
Noemia Teixeira de Siqueira-Filha http://orcid.org/0000-0003-0730-8561
Tom Wingfield http://orcid.org/0000-0001-8433-6887

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
