## [Reviewer comments · BMJ Open]

ARTICLE DETAILS

TITLE (PROVISIONAL)	Barriers and facilitators to accessing tuberculosis care in Nepal: a qualitative study to inform the design of a socioeconomic support intervention
AUTHORS	Dixit, Kritika; Biermann, Olivia; Rai, Bhola; Aryal, Tara; Mishra, Gokul; Teixeira de Siqueira-Filha, Noemia; Paudel, Puskar; Pandit, Ram; Sah, Manoj; Majhi, Govinda; Levy, Jens; Rest, Job; Gurung, Suman; Dhital, Raghu; Lönnroth, Knut; Squire, Bertie; Caws, Maxine; Sidney, Kristi; Wingfield, Tom

VERSION 1 – REVIEW

REVIEWER	Mia Bierbaum Macquarie University, Australian Institute of Health Innovation
REVIEW RETURNED	02-Mar-2021

GENERAL COMMENTS	This qualitative study aimed to determine the barriers and facilitators to accessing a TB support in Nepal. As part of a larger mixed method project seeking to implement a socioeconomic intervention for TB support, the findings from this study highlight key areas that will be targeted to improve the implementation of the intervention. This study is relevant and will inform the intervention of future interventions in LMICs. The study uses appropriate methods to validate the findings, such as member checking and use of multiple coders. The findings are comprehensively reported Informative title. Clear and informative abstract. A few suggestions for minor amendments: Abstract Intro - P3 line 9 – '10 million people developed TB' Is this globally? Clarify. Similarly, if three million were never diagnosed, should it read 'it is estimated that...' Abstract Methods – it could be clearer in the abstract whether the FGDs were a combination of the participants from different backgrounds, or whether you grouped TB patients, stakeholders and healthcare professionals together. Describe briefly the strategies you used to ensure validity of data e.g., member checking and multiple coders and a consensus-based coding/ development of a thematic framework. Abstract Results – without reading the rest of the manuscript, it is unclear that “vouchers and nutritional allowance to cover food and travel costs” relates to accessing TB support. Clarify that this is support for people with TB to access care and services. Abstract conclusion – reads easily and contextualises the results well.
---

	Background Comprehensive first paragraph. P6 Line 8 – as discussed for the abstract, given the 2.9 million people with undiagnosed TB, should the 10 million be referred to as an estimate? P8 line 35 – rephrase to highlight the important point that TB related costs amount to greater than 20% of a household’s annual income. Materials and methods Good use of COREQ checklist. Clear description of the study setting. Figure 1 – unclear what figure 1 adds to the manuscript. Could be removed, or a more informative description added. Sampling -P7 line 47 More detail needed. How were the participants with TB identified? Were their details gathered from the medical clinics? Supplementary file 1 and in text– add more detail about the community leaders/civil society organisation/community mobilisers/healthcare providers (these titles are vague on their own) – provide examples of the professional/community experiences of these individuals to justify why they were selected. P8 line 12 – this needs rephrasing - the FGDs didn’t all have 7 participants e.g. one had 6, one had 12. Describe this more clearly. Include the final numbers in each FGD in the results section. 55 individuals were invited to participate, but only 7 per FGD. What happened to the remaining 6 participants? Were they part of the pilot group? Analysis Clarify this section further. Did KD and TW independently code the same transcripts? How did you ensure they were using the same coding framework? You described that through consensus, they independently reviewed the coding and themes. Did they then recode the transcripts? Good technique to use multiple coders, just clarify the order of events here. Results P11 line 17 Briefly describe the number of participants, their age/gender/professional background here. The results are clear and logically presented. Good use of quotes to highlight key points. Discussion The discussion contextualises the results well by linking the findings to potential improvements and implementation strategies for interventions to increase access to TB support services. P19. Line 57 – awkward phrasing. Consider rewording. References are relevant. Many references missing the journal titles. If an article is sourced from a journal, the website is not required in the reference list citation, however, inclusion of the DOI is. Overall, this valuable piece of research was a pleasure to read.
--	--

REVIEWER	Lina Davies Forsman Karolinska Institute, Department of Medicine, Unit of Infectious Disease
REVIEW RETURNED	18-May-2021

GENERAL COMMENTS	The authors have performed an interesting and thorough study highlighting important aspects of TB care in Nepal. The article is well-written and clearly explained. I have some minor comments/questions In line 56 there is a quote of a person who had contacted 15 different pharmacies without getting the advice to test for TB. This is a major barrier to diagnosis. How can this be addressed? Furthermore, you state that your study only applies for the state funded TB care. How big proportion of TB care is private? If this is a big part it cannot be ignored for your implementation study to have an impact. Figure 3 in the method paper seems to be overlapping with the themes identified in the results (WHO). Where there any findings worth highlighting that did not fit in to your a priori knowledge? Is it an objective analysis of the data, if you had already decided which themes to look for?
--

VERSION 1 – AUTHOR RESPONSE

Reviewer: 1

Ms. Mia Bierbaum, Macquarie University

A few suggestions for minor amendments:

Reviewer 1, Comment 1: Abstract Intro - P3 line 9 – '10 million people developed TB' Is this globally? Clarify. Similarly, if three million were never diagnosed, should it read 'it is estimated that...'

Response: Thank you for your suggestion. This sentence has been removed from the abstract and amended in the main introduction section.

Reviewer 1, Comment 2: Abstract Methods – it could be clearer in the abstract whether the FGDs were a combination of the participants from different backgrounds, or whether you grouped TB patients, stakeholders and healthcare professionals together.

Response: We appreciate your suggestion and agree that it adds clarity in the sentence in the participants sub-heading in the abstract. The sentence is now updated as: *'Seven FGDs were conducted with 54 in-country stakeholders, grouped by stakeholders, including people with TB (n=21), community stakeholders (n=13), and multidisciplinary TB healthcare professionals (n=20) from the National TB Program.'*

Reviewer 1, Comment 3: Describe briefly the strategies you used to ensure validity of data e.g., member checking and multiple coders and a consensus-based coding/ development of a thematic framework.

Response: Thank you for the suggestion to elaborate on the methods. We have updated the methods section to address this as follows: *‘The data was managed in NVivo 12, coded by consensus, and analysed thematically.’*

We have also updated other sections of the manuscript to address this comment:

Strengths and Limitations bullet points: *‘The credibility and trustworthiness of the study was maintained through member checking, using multiple coders, conducting a consensus-based coding, recruiting local interviewers for data collection, performing triangulation and including a broad selection of multidisciplinary stakeholders to inform the study conclusion.’*

Methods, Data Collection section, Page 8, Lines 53-58: *‘We performed real-time member checking in each FGD by noting key points of the discussion, summarizing them on a wall chart, and clarifying their accuracy with the group.’*

Methods, Analysis section, Page 10, Lines 42-57: *‘The study used multiple coders, KD and TW, who familiarised themselves with the data through successive reading of transcripts. KD and TW separately generated the initial codes for each transcript before discussing and comparing the perception of understanding of the codes. The codes were updated through regular discussion as further data became available and collated following each successive FGD. To increase trustworthiness of the study, after all the transcripts were coded and analysed, KD and TW independently reviewed coding and themes and refined them through further discussion, triangulation, and consensus where necessary.’²¹*

Reviewer 1, Comment 4: Abstract Results – without reading the rest of the manuscript, it is unclear that “vouchers and nutritional allowance to cover food and travel costs” relates to accessing TB support. Clarify that this is support for people with TB to access care and services.

Response: Thank you for the opportunity to clarify this statement. We have now updated the sentence as *‘The perceived facilitators to accessing TB care and services were: enhanced championing and awareness-raising about TB and services; social protection including health insurance; cash, vouchers and/or nutritional allowance to cover food and travel costs; and psychosocial support and counseling integrated with existing adherence counseling from the National TB Program.’*

Reviewer 1, Comment 5: Background: Comprehensive first paragraph. P6 Line 8 – as discussed for the abstract, given the 2.9 million people with undiagnosed TB, should the 10 million be referred to as an estimate?

Response: Thank you for highlighting this sentence for clarification. We have now corrected it in the updated manuscript in Page 5, Line 8 as *'In 2019, an estimated 10 million became ill with TB, of whom 2.9 million were not notified or remained undiagnosed and untreated.'*

Reviewer 1, Comment 6: P8 line 35 – rephrase to highlight the important point that TB related costs amount to greater than 20% of a household's annual income.

Response: We have now rephrased this sentence as *'Despite free basic TB diagnostic tests, medicines and financial support for people with drug-resistant (DR-TB), approximately one in two people with TB face catastrophic costs (defined as the total TB-related costs equivalent to greater than 20% of a household's annual income) while accessing TB care in Nepal.'*

Reviewer 1, Comment 7: Figure 1 – unclear what figure 1 adds to the manuscript. Could be removed, or a more informative description added.

Response: We thank the reviewer for your comment. However, we would like to keep Figure 1 to support reader orientation to the study setting and sites. A more informative descriptive to the legend has now been added in the legend: *'Figure 1: The highlighted color represents the study districts in Nepal. Dhanusha, Mahottari and Chitwan are 'plains' or 'Terai' districts. Makwanpur is a hilly district. The district's data for population numbers and TB case notification rate highlights the burden of tuberculosis in each district (National TB Control Center Annual Report, 2018).'*

Reviewer 1, Comment 8: Sampling -P7 line 47: More detail needed. How were the participants with TB identified? Were their details gathered from the medical clinics?

Response: We have added a sentence regarding the selection of participants in the manuscript in Page 7, Lines 5-10 as *'The list of people with TB, including their demographics were gathered from the IMPACT TB database or registers of the health clinics in each district.'*

Reviewer 1, Comment 9: Supplementary file 1 and in text– add more detail about the community leaders/civil society organisation/community mobilisers/healthcare providers (these titles are vague on their own) – provide examples of the professional/community experiences of these individuals to justify why they were selected.

Response: Thank you for drawing our attention to this. The sentence has been updated to better clarify the characteristics of the study participants in Page 7, Lines 10-27: *'Community stakeholders were community leaders or those working in civil society and were selected based on their in-depth knowledge on the local context and experiences of working with the communities, preferable in health'*

programs. TB healthcare professionals, such as those working with the NTP or TB-focused NGOs, had several years' experience in planning, designing and implementing NTP activities. Community volunteers or mobilizers were people working with the IMPACT TB project, who have first-hand experience in screening symptoms of TB and supporting people with TB to adhere to and complete their treatment. These participants were selected based on the expertise in delivering community programs and to bring diverse perception of the stakeholders in the study'.

Reviewer 1, Comment 10: P8 line 12 – this needs rephrasing - the FGDs didn't all have 7 participants e.g. one had 6, one had 12. Describe this more clearly. Include the final numbers in each FGD in the results section. 55 individuals were invited to participate, but only 7 per FGD. What happened to the remaining 6 participants? Were they part of the pilot group?

Response: We have now clarified this by moving the supplementary table detailing FGDs conducted and their participants to the main text. In addition, we have updated the Methods, Data collection section, Page 7, Lines 31-39: *'Seven participants were invited to each of the seven FGDs with the exception of the TB healthcare professional FGD, which consisted of 12 participants. This related to the logistical challenges of organizing more than one FGD with this group due to their working hours and time constraints coupled with the aim of representation from the public, private, and NGO sectors of TB healthcare.'*

We have also mentioned that some of the participants with TB were used as a pilot group, Methods, Data Collection section, Page 8, Lines 17-21: *'The topic guide was piloted with a group of seven female and male participants with TB resulting in minor refinements to the FGD structure and delivery techniques.'*

Reviewer 1, Comment 11: Analysis: Clarify this section further. Did KD and TW independently code the same transcripts? How did you ensure they were using the same coding framework? You described that through consensus, they independently reviewed the coding and themes. Did they then recode the transcripts? Good technique to use multiple coders, just clarify the order of events here.

Response: Thank you for the opportunity to clarify the analysis that we performed in the study. This text has been update to read *'The study used multiple coders, KD and TW, who familiarised themselves with the data through successive reading of transcripts. KD and TW separately generated the initial codes for each transcript before discussing and comparing the perception of understanding of the codes. The codes were updated through regular discussion as further data became available and collated following each successive FGD. To increase trustworthiness of the study, after all the transcripts were coded and analysed, KD and TW independently reviewed coding and themes and refined them through further discussion, triangulation, and consensus where necessary.'*²¹.

Reviewer 1, Comment 12: Results: P11 line 17. Briefly describe the number of participants, their age/gender/professional background here.

Response: We have now moved the supplementary table detailing the participants of the FGDs to the main document in Table 1 (Page 9, Lines 13-60; Page 10, Lines 3-39), and rephrased the relevant text to be clearer as follows:

'In all the FGDs, there were seven participants, except for the FGD with community leaders (n=6), FGD with TB healthcare professionals (n=12) and FGD with Community Mobilizers (n=8). Of the participants, 38/54 (70%) were male and the average age was 42 years.'

Since purposive sampling was used, we describe the participants' characteristics in the methods section.

Reviewer 1, Comment 13: P19. Line 57 – awkward phrasing. Consider rewording.

Response: Thank you for this suggestion. We have since updated the text in Page 20, Lines 28-34 in the discussion to read: *'Previously, similar campaigns focused on TB awareness through door-to-door visits, health promotion at health facilities, or educational outreach into communities. Such campaigns were perceived to increase knowledge on TB, advocate for free TB services, and empower communities to make informed choices.'*

Reviewer 1, Comment 14: References: References are relevant. Many references missing the journal titles. If an article is sourced from a journal, the website is not required in the reference list citation, however, inclusion of the DOI is.

Response: We apologise for the missing elements in the references and have carefully updated the journal title and re-proofed the manuscript to correct them throughout.

Reviewer: 2

Dr. Lina Davies Forsman, Karolinska Institute, Karolinska University Hospital Solna

Reviewer 2, Comment 1: In line 56 there is a quote of a person who had contacted 15 different pharmacies without getting the advice to test for TB. This is a major barrier to diagnosis. How can this be addressed?

Response: Thank you for highlighting this important issue. In the Discussion section, Page 22, Lines 14-22, we have included the following sentence to address the query: *'Studies in India and Vietnam have demonstrated enhanced engagement with private pharmacies and medical practitioners by providing them subsidies directly or through intermediary agencies for every notified case.^{55,56} This could be a potential strategy to improve access to TB care in Nepal, where approximately 20% people with TB receive paid treatment from the private sector.⁵⁷*

Reviewer 2, Comment 2: Furthermore, you state that your study only applies for the state funded TB care. How big proportion of TB care is private? If this is a big part it cannot be ignored for your implementation study to have an impact.

Response: We appreciate your comment regarding the importance of private sector TB care, which is a significant component of TB care in many high burden countries including Nepal. We have addressed this issue in the discussion section and have aligned it with your previous comment in Page 22, Lines 14-22 as: *‘Studies in India and Vietnam demonstrated enhanced engagement with private pharmacies and medical practitioners by providing them subsidies directly or through intermediary agencies for every notified cases.^{55,56} This could be a potential strategy to improve access to TB care in Nepal, where approximately 20% people with TB receive paid treatment from the private sector.⁵⁷*

Reviewer 2, Comment 3: Figure 3 in the method paper seems to be overlapping with the themes identified in the results (WHO). Where there any findings worth highlighting that did not fit in to your a priori knowledge? Is it an objective analysis of the data, if you had already decided which themes to look for?

Response: Thank you for this valuable comment. Figure 3 of the study protocol paper published in Wellcome Open Research (<https://wellcomeopenresearch.org/articles/5-19>) shows a conceptual framework of five categories that may affect medication adherence and be amenable to intervention. The barriers and facilitators identified in that conceptual framework are broad and not country-specific. Collectively, the research team felt the framework and its categories best captured the issues we had seen in practice in the field relating to access to TB services.

In our analysis, we adapted that framework and chose to focus on the following categories: “TB, health, and basic education”, “Social protection and nutrition”, and “Psychosocial” as these were modifiable by our household level (rather than health system level) intervention. During analysis of our focus group discussion data, we mapped our findings to these four broad categories adapted from the WHO adherence framework.

It is inevitable that some of the broader barriers and facilitators that impact upon adherence would be elicited in the Nepali context. However, our analysis allows us to identify those barriers most pertinent to the Nepali context and, in the wider programme of research within which this study was housed, develop focused socioeconomic support interventions to address them.

VERSION 2 – REVIEW

REVIEWER	Mia Bierbaum Macquarie University, Australian Institute of Health Innovation
REVIEW RETURNED	21-Jun-2021
GENERAL COMMENTS	The comments have all been attended to, and the paper now reads well.